# Modelling BK Polyomavirus dissemination and cytopathology using polarized human renal tubule epithelial cells

Elias Myrvoll Lorentzen[1,2], Stian Henriksen[1,2], Christine Hanssen Rinaldo [1,2] *

1 Department of Microbiology and Infection Control, University Hospital of North Norway, Tromsø, Norway,
2 Metabolic and Renal Research Group, Department of Clinical Medicine, UiT The Arctic University of Norway, Tromsø, Norway

* christine.rinaldo@unn.no

## Abstract

Most humans have a lifelong imperceptible BK Polyomavirus (BKPyV) infection in epithelial cells lining the reno-urinary tract. In kidney transplant recipients, unrestricted high-level replication of donor-derived BKPyV in the allograft underlies polyomavirus-associated nephropathy, a condition with massive epithelial cell loss and inflammation causing premature allograft failure. There is limited understanding on how BKPyV disseminates throughout the reno-urinary tract and sometimes causes kidney damage. Tubule epithelial cells are tightly connected and have unique apical and basolateral membrane domains with highly specialized functions but all *in vitro* BKPyV studies have been performed in non-polarized cells. We therefore generated a polarized cell model of primary renal proximal tubule epithelial cells (RPTECs) and characterized BKPyV entry and release. After 8 days on permeable inserts, RPTECs demonstrated apico-basal polarity. BKPyV entry was most efficient via the apical membrane, that *in vivo* faces the tubular lumen, and depended on sialic acids. Progeny release started between 48 and 58 hours post-infection (hpi), and was exclusively detected in the apical compartment. From 72 hpi, cell lysis and detachment gradually increased but cells were mainly shed by extrusion and the barrier function was therefore maintained. The decoy-like cells were BKPyV infected and could transmit BKPyV to uninfected cells. By 120 hpi, the epithelial barrier was disrupted by severe cytopathic effects, and BKPyV entered the basolateral compartment mimicking the interstitial space. Addition of BKPyV-specific neutralizing antibodies to this compartment inhibited new infections. Taken together, we propose that during *in vivo* low-level BKPyV replication, BKPyV disseminates inside the tubular system, thereby causing minimal damage and delaying immune detection. However, in kidney transplant recipients lacking a well-functioning immune system, replication in the allograft will progress and eventually cause denudation of the basement membrane, leading to an increased number of decoy cells, high-level BKPyV-DNAuria and DNAemia, the latter a marker of allograft damage.

**Data Availability Statement:** All relevant data are within the manuscript and its supporting Information files.

**Funding:** This work was supported by a grant from the Northern Norway Regional Health Authority – project number HNF1571-21 to CHR. The funders had no role in study design, data collection and analysis, decision to publish, or preparation of the manuscript.

**Competing interests:** The authors have declared that no competing interests exist.

## Author summary

BKPyV causes polyomavirus-associated nephropathy, a severe condition affecting kidney transplant recipients. Besides, BKPyV is commonly detected in urine of healthy individuals. The renal tubules are lined by polarized epithelial cells that form a physical barrier with specialized functions. This is the first *in vitro* study of BKPyV replication in polarized tubule epithelial cells, a model reflecting the renal tubule anatomy. Cytopathic effects described in patients with polyomavirus-associated nephropathy, such as cell lysis and cell shedding, were recreated. Moreover, we demonstrate that BKPyV enters epithelial cells from the apical side and that viral progeny and infected cells are released into the apical compartment, which is mimicking the tubular lumen, without disrupting the epithelial barrier. Eventually the barrier was disrupted and BKPyV leaked into the basolateral compartment, mimicking the interstitial space. Our results suggest that in healthy individuals, viral progeny disseminates in the tubular fluid, thus delaying immune detection. In kidney transplant recipients with a suppressed immune system, BKPyV replication in the allograft may progress and cause massive cell loss and leakage of BKPyV-DNA into blood. Summarized, we have established a useful model to study renal BKPyV infection and used it to deepen our understanding of BKPyV dissemination and cytopathic effects.

## Introduction

BK Polyomavirus (BKPyV), one of the 13 known human polyomaviruses [1,2], infects more than 90% of the population worldwide [3]. BKPyV persists in epithelial cells of the reno-urinary tract and is intermittently shed in the urine of healthy individuals without causing symptoms [4]. In immunosuppressed individuals, mainly kidney transplant and allogeneic stem cell transplant recipients, unrestricted BKPyV replication causes polyomavirus-associated nephropathy (PyVAN) [5] and polyomavirus-associated hemorrhagic cystitis [6], respectively. Moreover, some PyVAN patients develop bladder cancer years later [7–9]. PyVAN affects 1–15% of kidney transplant patients. The early phase is characterized by uncontrolled BKPyV replication in tubule epithelial cells in isolated nephrons of the allograft, resulting in BKPyV viruria with little impact on renal function. Somehow BKPyV disseminates to multiple nephrons, which is causing high-level viruria, urinary decoy cells and high-level BKPyV DNAemia [10–14]. Finally, interstitial and tubular inflammatory infiltrates become prominent, contributing to the declining allograft function. As no effective anti-viral therapies are available and reduced immunosuppression is the only treatment option, PyVAN is an important cause of reduced allograft function and premature allograft loss [5].

The replication cycle of BKPyV has been studied in various non-polarized cell cultures such as African green monkey kidney cell lines [15] and more recently in primary human renal proximal tubule epithelial cells (RPTECs) [16]. However, these cell cultures differ greatly from epithelial cells *in vivo*, where apico-basal polarity yields two unique membrane domains which are essential for the cell shape and function [17]. Apico-basal polarity can have important influence on the viral replication cycle [18]. For instance, polarized distribution of viral receptors can restrict viral entry to one membrane domain while directional protein sorting can lead to directional progeny release [18]. The influence of apico-basal polarity on BKPyV entry and release is not characterized and we therefore have limited understanding of how BKPyV disseminate from tubule epithelial cells and spread throughout the reno-urinary tract and sometimes causes kidney damage. In an effort to answer these questions, we established an *in*

*vitro* model of polarized human RPTECs and used this model to characterize major steps of the BKPyV replication cycle.

## Results

### Renal proximal tubule epithelial cells develop a polarized morphology and functionality

To examine if virus entry and release is polarized, we needed access to both membrane domains. We therefore chose to utilize permeable cell culture inserts (Fig 1A), previously used for polarization of renal epithelial cells [19–22]. To examine if RPTECs developed a polarized morphology, they were cultured on Falcon-inserts with pore size 1.0 μm. At 8 days post-seeding (dps), immunofluorescence staining and confocal microscopy demonstrated hallmarks of polarized epithelial cells such as basolateral distribution of sodium-potassium ATPase (Na/K-ATPase) (Fig 1B), apical primary cilia shown by acetylated α-tubulin (Fig 1B) and intercellular tight junctions represented by zonula occludens-1 (ZO-1) (Fig 1C).

In the nephron, the epithelium in the proximal tubule is known to be leakier than the epithelium in the distal parts of the nephron [23,24]. To investigate the integrity of the tight junctions in polarized RPTECs, we measured the transepithelial electrical resistance (TEER). It gradually increased during culture and plateaued at 20–30 $\Omega * cm^2$ from 8 dps (Fig 1D). The barrier function was further examined by measuring the diffusion of FITC-dextran across the cell layer. Compared to non-polarized RPTECs (at 2–3 dps), diffusion across polarized RPTECs (at 8–10 dps) was reduced by 65% (Fig 1E).

Epithelial cells of the proximal tubule have several drug transporters, including the P-glycoprotein (P-gp) efflux pump [25]. To evaluate the functionality of P-gp, the cell-permeant calcein AM and the P-gp inhibitor Psc-833 can be used [26]. If P-gp is inhibited, calcein AM accumulates intracellularly and is hydrolyzed into fluorescent calcein. Polarized RPTECs were exposed to calcein AM in the presence or absence of Psc-833 before fluorescence was measured. Inhibition of P-gp increased the intracellular fluorescence by approximately 70% (Fig 1F), confirming that polarized RPTECs have an active P-gp efflux pump. Transwell-inserts with pore size 0.4 μm yielded results similar to pore size 1.0 μm for polarity markers, FITC-diffusion and P-gp activity (S1A–S1E Fig). Falcon-inserts with pore size 3.0 μm were unsuitable, as RPTECs migrated through the pores (S1F Fig).

In summary, our morphological and functional assessment confirmed that RPTECs cultured on inserts develop a polarized morphology and exhibit functions similar to epithelial cells in the proximal tubule. The subsequent virus experiments were performed from 8 dps, when TEER values plateaued.

### Polarized RPTECs support BKPyV replication

We next investigated if the polarized RPTECs were permissive for BKPyV. Two hours after apical infection, confocal microscopy demonstrated punctate Vp1-staining (Fig 2A), indicating binding and internalization of BKPyV. Transmission electron microscopy (TEM) confirmed binding of BKPyV to the apical membrane (Fig 2B) and a polarized morphology with microvilli, tight junctions and basal labyrinths (Fig 2B). At 72 hours post-infection (hpi), immunofluorescence staining (Fig 2C) and immunoblot (Fig 2D) showed expression of BKPyV proteins, while confocal microscopy demonstrated enlarged nuclei with inclusions (Fig 2E).

In summary, we found that polarized RPTECs supported BKPyV replication and may therefore be a suitable model for studying entry and release of BKPyV.

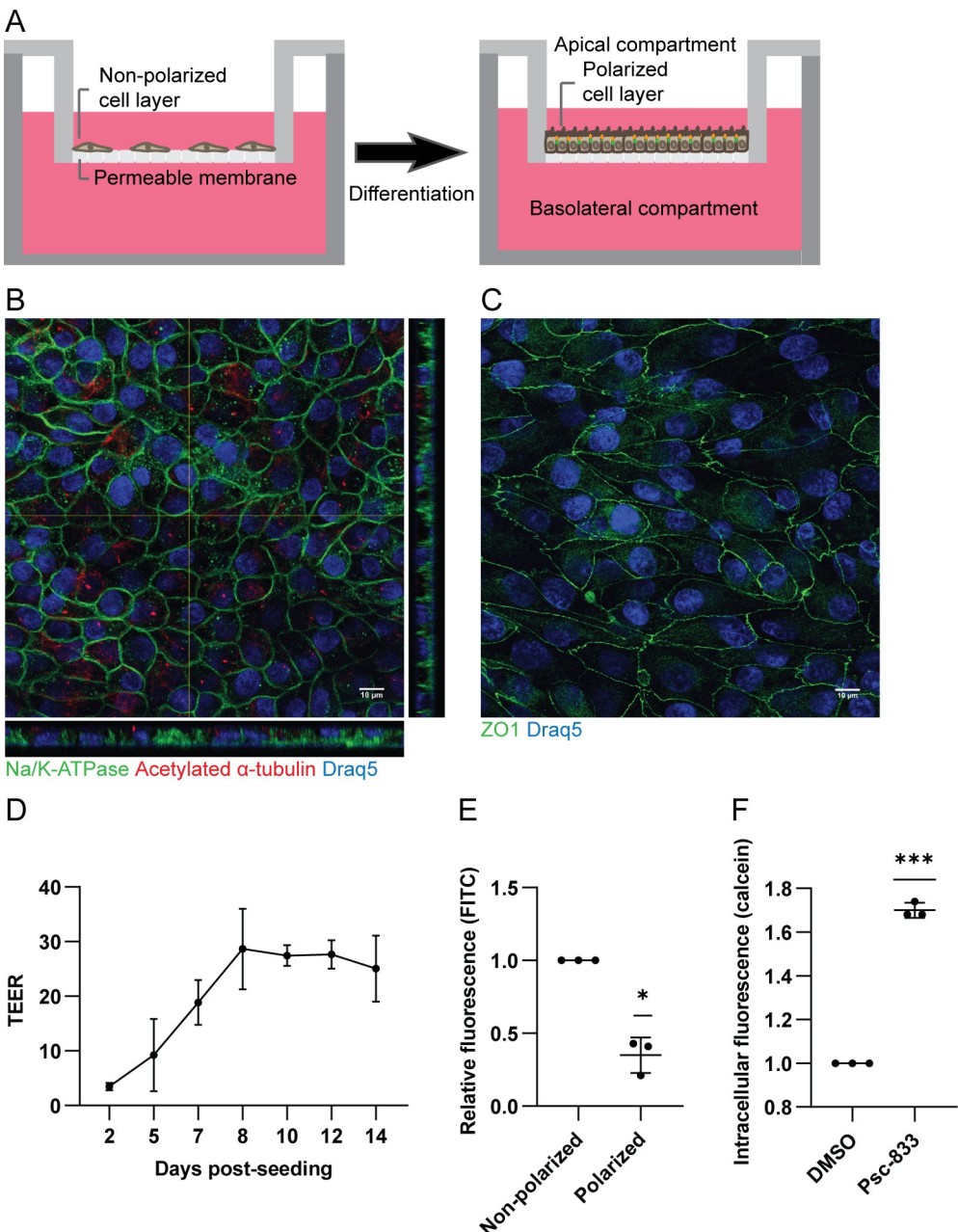

**Fig 1. RPTECs develop a polarized morphology and functionality on cell culture inserts.** (A) Scheme of cells cultured on a permeable cell culture insert. (B and C) At 8 days post-seeding (dps), RPTECs on Falcon-inserts were fixed and stained for: (B) Na/K-ATPase (green) and acetylated α-tubulin (red), and (C) ZO-1 (green). Nuclei were stained with Draq5 (blue). Images are representative images from at least three independent experiments. Scale bar 10 μm. (D) Transepithelial electrical resistance (TEER) values of RPTECs at 2 to 14 dps. Data is derived from two to four biological replicates per timepoint. Data are shown as means and error bars represent ± standard deviation (SD). (E) Diffusion of FITC-dextran from the apical to the basolateral compartment across polarized and non-polarized RPTECs. Data is normalized to the non-polarized control, n = 3 and error bars represent ± SD. * = P < 0.05, one sample *t* test. (F) Accumulation of intracellular calcein AM in the presence or absence of the P-gp inhibitor Psc-833, quantified by fluorescence measurement using a plate reader. Data is normalized to the untreated control, n = 3 and error bars represent ± SD. *** = P < 0.001, one sample *t* test.

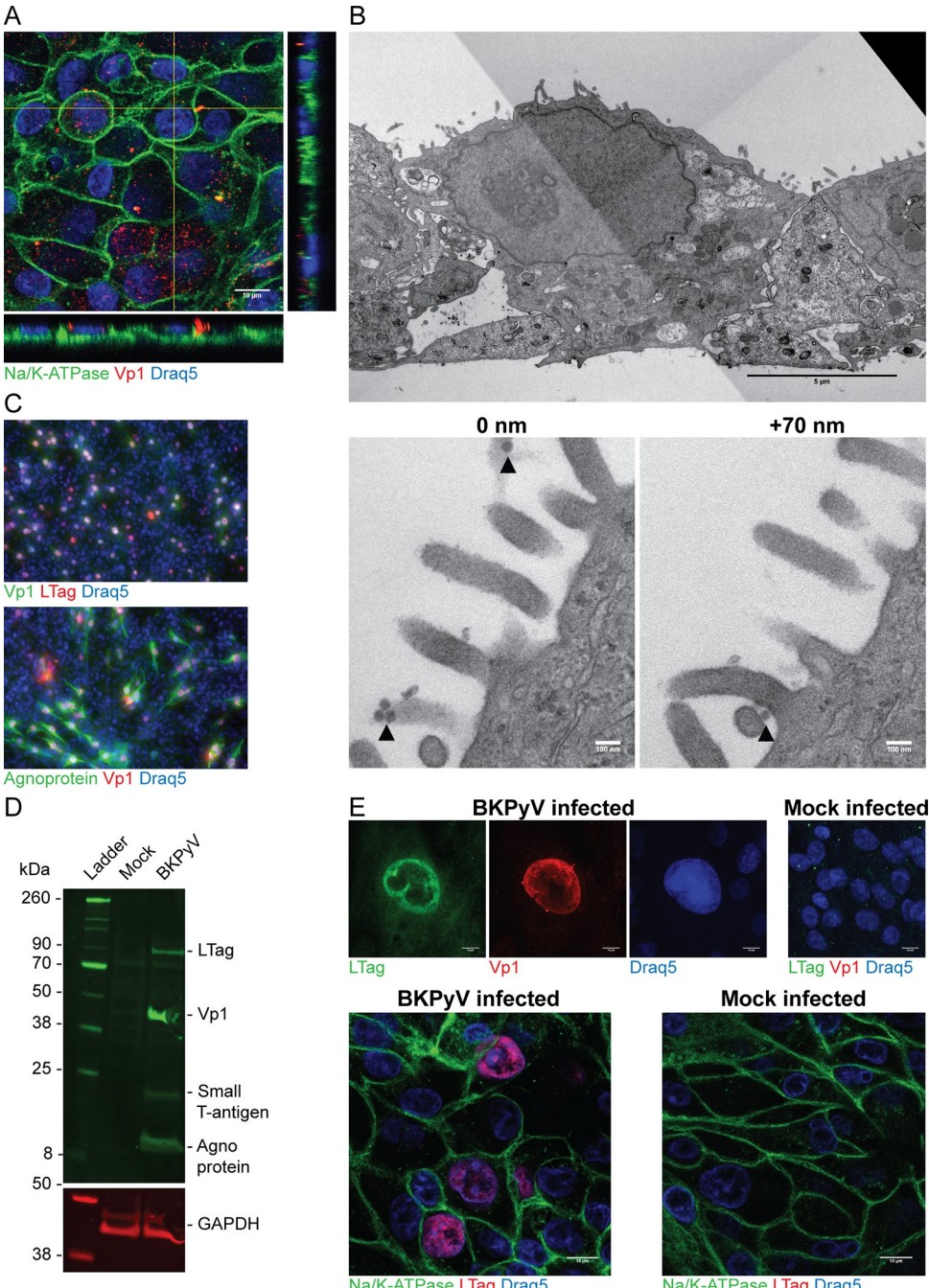

**Fig 2. Polarized RPTECs support BKPyV infection.** (A) Binding and internalization of BKPyV (MOI 0.1) in polarized RPTECs at 2 hours post infection (hpi) demonstrated by immunofluorescence staining against Vp1 (4942) (red) and Na/K-ATPase (green). Scale bar is 10 μm. (B) Transmission electron micrographs of polarized RPTECs that have been inoculated with BKPyV for 2 hours. Top image is an overview image with scale bar 5 μm. Bottom images are representative images from 70 nm serial sections of virions on the cell surface (black arrowhead). Scale bar 100 nm. Productive BKPyV infection in polarized RPTECs at 3 days post infection (dpi) with BKPyV (MOI 1) is demonstrated by: (C) Immunofluorescence staining against Vp1 (rabbit serum) (green) and LTag (Pab416) (red) (top image) or agnoprotein (green) and Vp1 (4942) (red) (bottom image). (D) Western blot using rabbit serums against N-terminal LTag, Vp1 and agnoprotein. Lysates of mock infected cells were used as negative control and a GAPDH antibody was used as a loading control. (E) Confocal microscopy images of BKPyV infected cells. Top images are stained with rabbit serum against N-terminal LTag (green) and an antibody against Vp1 (4942) (red). Bottom images are stained for Na/K-ATPase (green) and LTag (Pab416) (red). Mock infected cells are included as a negative control. Scale bar 10 μm. In (A), (C) and (E), nuclei are stained with Draq5 (blue) and representative images from three independent experiments are shown.

## BKPyV preferentially enters polarized RPTECs through the apical membrane

It is not known which compartment BKPyV must access to infect renal tubule epithelial cells *in vivo*. To shed light on this, we examined the entry of BKPyV in our polarized model. To ensure that BKPyV could traverse the insert, we first determined the diffusion of BKPyV through empty inserts. Approximately 19% of applied virus diffused across Falcon-inserts (1.0 μm) while only 5% diffused across the Transwell-inserts (0.4 μm). Based on this, the Falcon-inserts and a 5.3x higher BKPyV concentration for basolateral infections were used for subsequent experiments.

We infected polarized RPTECs via the apical membrane or the basolateral membrane and determined infectivity by immunofluorescence staining. Basolateral infection yielded 71% fewer infected cells than apical infection (Figs 3A and S2A). We hypothesized that this was caused by poorer binding of BKPyV to the basolateral membrane. To examine this, we repeated the previous experiment except that we performed immunofluorescence staining at 2 hpi and compared the intensity of the Vp1-signal. Apical infection yielded a significantly stronger Vp1-signal (Fig 3B and 3C), indicating that more Vp1 bound to the apical membrane than the basolateral membrane. As a control, we infected RPTECs at 2 dps i.e. prior to polarization, and found no difference in infectivity between apical and basolateral infection (Figs 3D and S2B). This confirms that apico-basal polarity is necessary for preferential apical entry.

It has previously been shown that BKPyV can utilize sialic acids and gangliosides on non-polarized Vero cells as receptors [27–31]. To assess if sialic acids are necessary for BKPyV infection in polarized RPTECs, we performed a neuraminidase pre-treatment. This reduced infectivity by 90% (Figs 3E and S2C), confirming that sialic acids are indispensable for BKPyV infection in polarized RPTECs.

In lack of suitable antibodies against gangliosides, Texas-Red conjugated wheat-germ-agglutinin (WGA) was used to examine the distribution of sialic acids. WGA-staining was almost exclusively seen on the apical membrane (Fig 3F). In non-polarized RPTECs, both apical and basolateral application of WGA yielded visible staining (S2D Fig).

We conclude that BKPyV mainly enters RPTECs through the apical membrane, possibly because of more sialic acids at the apical membrane. All subsequent infections were done via the apical compartment.

## BKPyV is mainly released into the apical compartment

*In vivo*, the direction of virus release has important consequences as it influences if viruses disseminate systemically or cause local infection [18]. For viruses infecting tubule epithelial cells, apical release will result in viruria while basolateral release will result in virus in the interstitial space and potentially viremia. To investigate if BKPyV undergo directional release, we infected polarized RPTECs (MOI 0.3), sampled the supernatants before removal of the inoculum at 2 hpi and at several later timepoints up to 120 hpi and analyzed BKPyV-DNA loads. At 2 hpi only 0.4% of the extracellular BKPyV-DNA load was detected in the basolateral compartment (S3A Fig), confirming that the epithelial cells formed a tight barrier. At 48 and 58 hpi, the extracellular BKPyV-DNA load in the apical compartment had increased by 0.7 log and 1 log from input, i.e. the inoculum left after washing (Fig 4A), suggesting that progeny release had started. The apical BKPyV-DNA load increased up to the last timepoint, 120 hpi, at which a 3.1 log increase was found (Fig 4A). Up to 58 hpi, the BKPyV-DNA load in the basolateral compartment was 2.4 to 2.8 log lower than in the apical compartment. The difference decreased to 1.3 log at 120 hpi. BKPyV infection with MOI 3 and 30 yielded similar results,

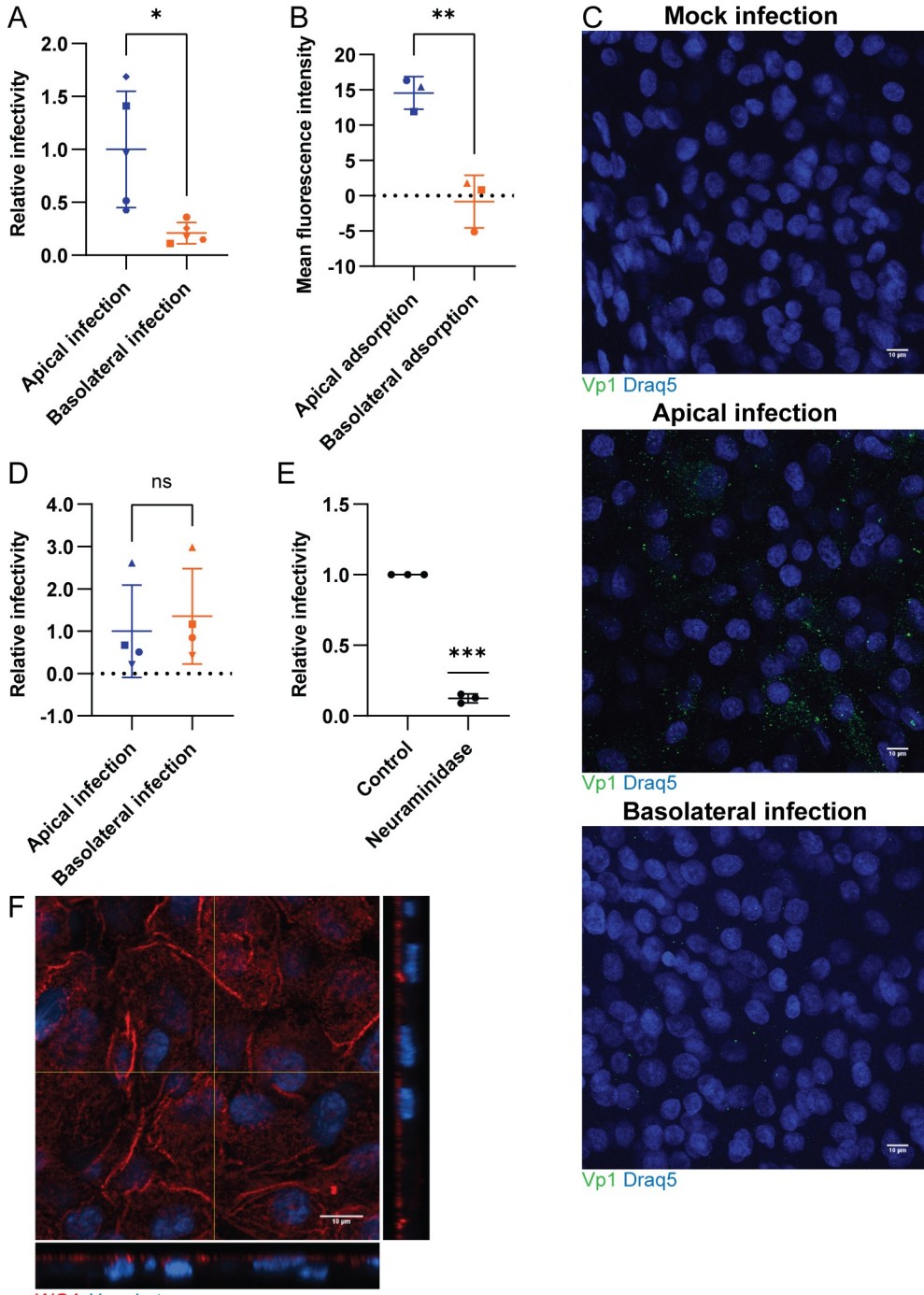

**Fig 3. BKPyV preferentially enters polarized RPTECs via the apical membrane.** (A) BKPyV infectivity following apical and basolateral infection. Apical infection was performed with MOI 0.1 while basolateral infection was done with 5.3x more virus. Data represents the number of infected cells based on immunofluorescence staining for Vp1 (4942) and agnoprotein at 3 dpi and is presented as relative infectivity normalized to the mean number of infected cells for apical infection. n = 5 and error bars represent ± SD. * = P < 0.05, two-tailed *t* test (B) Detection of BKPyV at 2 hours after apical and basolateral infection, respectively. Immunofluorescence staining for Vp1 (4942) was performed and followed by confocal microscopy and acquisition of z-stacks. Vp1-staining intensity was measured in sum z-projections and is represented as mean fluorescence intensity. Z-stacks of mock infected cells were used as a negative control and subtracted as background. Error bars represent ± SD and n = 3. ** = P < 0.01, two-tailed *t* test. (C) Representative z-slices from (B) stained for Vp1 (4942) (green) and Draq5 (blue). Scale bar 10 μm. (D) BKPyV infectivity in non-polarized RPTECs following apical or basolateral infection at 2 dps. Apical infection was performed

with approximately MOI 0.1 while basolateral infection was done with 5.3x more virus. Data represents the number of infected cells based on immunofluorescence staining for Vp1 (4942) and agnoprotein at 3 dpi and is presented as relative infectivity normalized to the mean number of infected cells for apical infection. n = 4 and error bars represent ± SD. ns = P > 0.05, two-tailed *t* test (E) BKPyV infectivity after neuraminidase-pretreatment. Data represents the number of infected cells based on immunofluorescence staining for Vp1 (4942) and agnoprotein at 3 dpi and is presented as relative infectivity normalized to the untreated control. n = 3 and error bars represent ± SD. *** = P < 0.001, one sample *t* test. (F) Representative apical z-slice from a z-stack of polarized RPTECs stained with Texas Red conjugated wheat germ agglutinin (red) and Hoechst (blue). Scale bar 10 μm.

except that the basolateral BKPyV-DNA load increased slightly earlier (S3B and S3C Fig). Of note, at 120 hpi, we observed considerable cytopathic effects (CPE).

Next, we examined if the extracellular BKPyV-DNA corresponded to infectious BKPyV by inoculating supernatants onto non-polarized RPTECs. Up to 72 hpi, virus was exclusively found in supernatants from the apical compartment (Fig 4B). In supernatants from the baso-lateral compartment, virus was detected at 120 hpi (Fig 4B), coincident with the observed CPE. Notably, the infectious BKPyV load was still about 2 log lower than in the apical supernatants.

We conclude that BKPyV is preferentially released into the apical compartment and that basolateral BKPyV represents leakage from the apical compartment. Detection of apical prog-eny release between 48 and 58 hpi, suggests that the BKPyV replication cycle is of similar length as in non-polarized RPTECs [32].

## BKPyV replication causes late cell death in polarized RPTECs

Due to the detection of CPE, we decided to examine the morphology of cells throughout BKPyV infection (MOI 3.0). Cytoplasmic vacuolization, cell rounding and loss of the cobble-stone-pattern emerged at 72 hpi, increased over time and was most evident at 120 hpi (Fig 5A). We also noted that a reduced infectious dose (MOI 1 or 0.3) gave less CPE (results not shown).

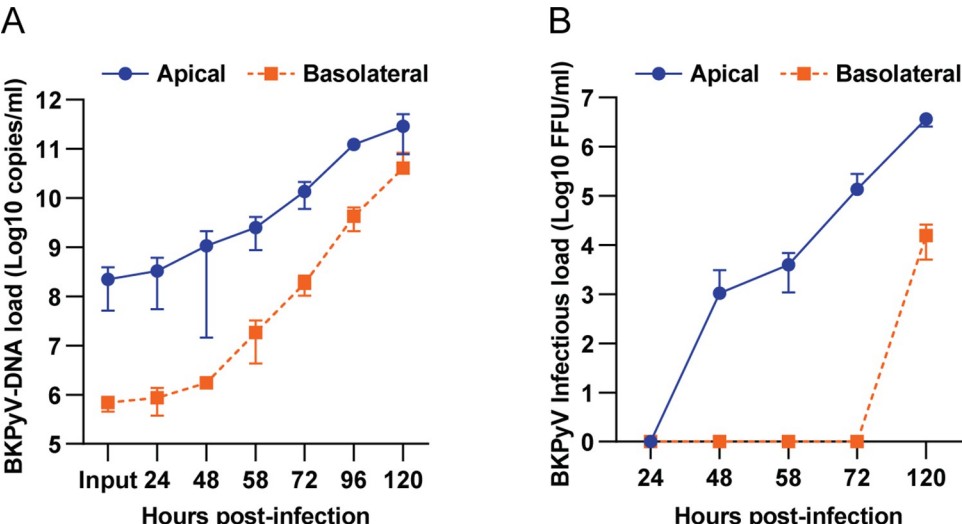

**Fig 4. BKPyV is mainly released into the apical compartment.** (A) Polarized RPTECs were apically infected (MOI 0.3) and supernatants were collected at the indicated timepoints for BKPyV-DNA load (log 10 copies/ml) determination by qPCR. Data was generated from at least three independent experiments, except the 96 hpi timepoint which was derived from two independent experiments. (B) Polarized RPTECs were apically infected (MOI 1) and supernatants were collected at the indicated timepoints for determination of BKPyV infectious load (log 10 FFU/ml) by infectivity assay. Infectious load at 24 hpi was defined as input and subtracted as background. Data was generated from six independent experiments. Error bars represent ± SD for (A) and (B).

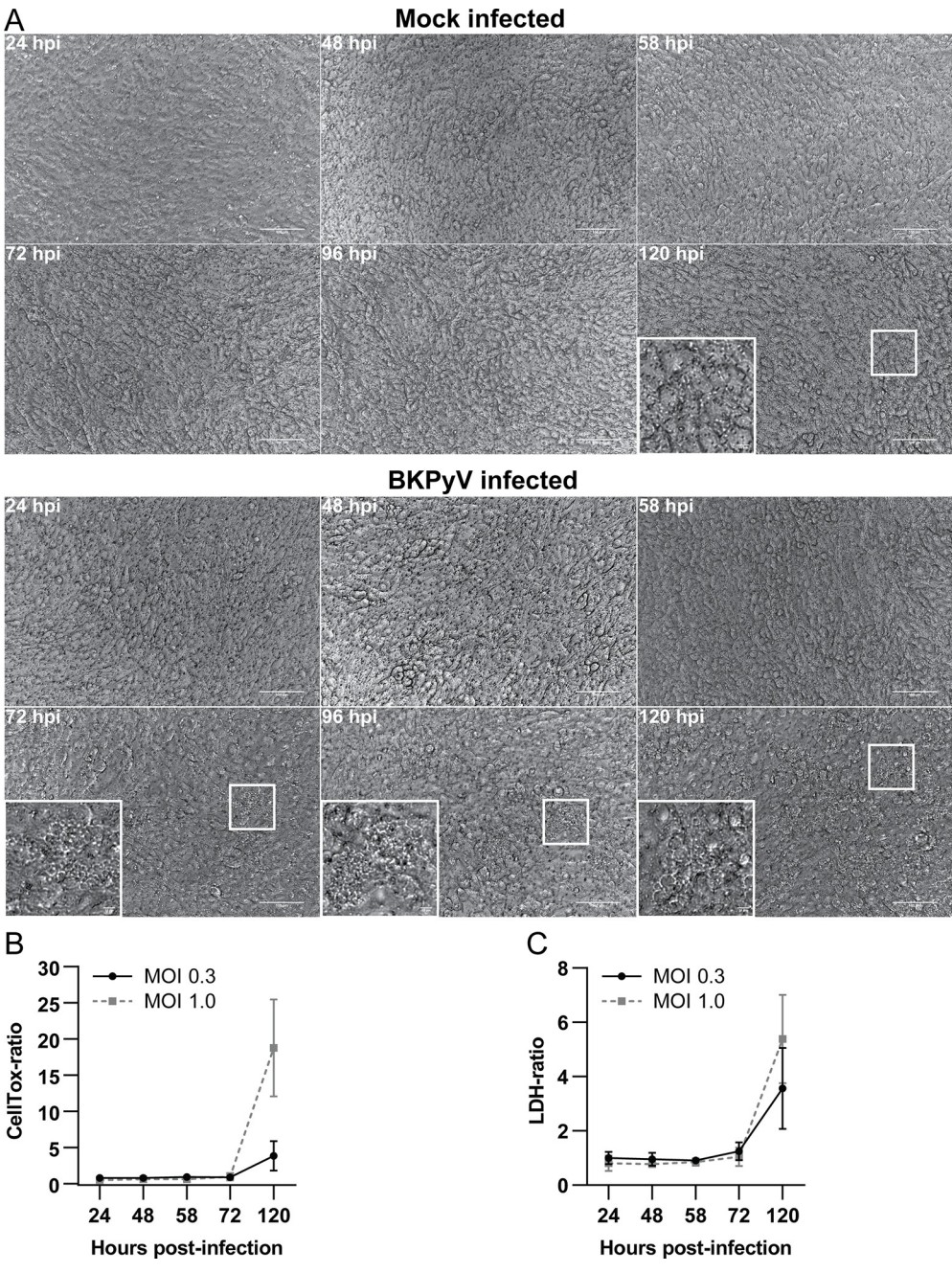

**Fig 5. BKPyV induced cytopathic effects and cell death in polarized RPTECs.** (A) Phase-contrast images of mock infected and BKPyV infected RPTECs (MOI 3) from 24 to 120 hpi. Representative images from two independent experiments are shown. Scale bar 100 μm. (B) Widefield microscopy of mock infected and BKPyV infected RPTECs incubated with CellTox dye. Data is presented as CellTox-ratio (mean total fluorescence from infected inserts/mean total fluorescence from mock infected inserts). Error bars represent ± SD and data is derived from at least three independent experiments. (C) Release of LDH into apical supernatants as measured by Promega LDH-Glo Cytotoxicity assay. Data is presented as LDH-ratio (infected-RLU/mock-RLU). Error bars represent ± SD and data is derived from at least three independent experiments.

Studies have suggested that BKPyV is released after host cell necrosis [10,11,33]. A common feature of cell death with necrotic morphology is permanent plasma membrane permeabilization [34]. We therefore examined the plasma membrane integrity of mock infected and

BKPyV infected RPTECs from 24 to 120 hpi by measuring CellTox-fluorescence and LDH release. Up to 72 hpi, BKPyV caused no increase in CellTox-fluorescence, but at 120 hpi, a 5- and 20-fold increase was found with BKPyV MOI 0.3 and MOI 1, respectively (Fig 5B) indicating host cell lysis. Similarly, no increase in extracellular LDH was observed up to 72 hpi, while at 120 hpi a 4- and 6-fold increase was detected for MOI 0.3 and MOI 1, respectively (Fig 5C). BKPyV infections with higher MOI (3 and 30) caused earlier and more prominent increase in CellTox-fluorescence and LDH release (S3D and S3E Fig).

Next, we investigated the plasma membrane integrity of individual cells by utilizing CellTox dye and live-cell imaging (Fig 6A and S1–S2 Video). An increase in CellTox-positive cells was exclusively observed in the infected inserts. The first increase was detected at 72 hpi with a mean of 32 new CellTox-positive cells per image, representing a 2.8-fold increase. The increase continued up to 120 hpi with 336 new CellTox-positive cells, representing a 30-fold increase (Fig 6A and 6B and S1 Video). Additionally, we observed cell ballooning and lysis in real-time (S1 Video). In contrast, the number of CellTox-positive cells consistently decreased throughout imaging for mock infected inserts (Fig 6A and 6B and S2 Video).

As BKPyV infection leads to widespread cell death, we investigated if this affected the barrier function. Up to 3 days post-infection (dpi), we detected a slight increase in TEER for BKPyV infected RPTECs. However, at 5 dpi we detected a negative trend in TEER with a 14% and 34% reduction for MOI 1 and MOI 10, progressing to a 20% and 50% reduction at 7 dpi, respectively (Fig 6C), indicating a disrupted barrier function.

We conclude that BKPyV induces CPE and lytic cell death in polarized RPTECs, but that this mainly occurs from 3 dpi. Despite CPE and increasing cell death, we first detected a downward trend in TEER at 5 dpi, allowing leakage of BKPyV from the apical to the basolateral compartment.

## BKPyV replication leads to extensive cell detachment

Many viruses are known to cause detachment of infected cells [35–37], including BKPyV as urinary shedding of infected epithelial cells, i.e decoy cells [38,39], is commonly observed in PyVAN patients [11,14]. Although a well-known phenomenon, this feature of BKPyV infection has not been studied *in vitro*.

First, we harvested supernatants from mock infected and BKPyV infected RPTECs (MOI 1) at 5 dpi and imaged them for detached cells. All supernatants contained detached cells, but the infected inserts yielded markedly more cells. Addition of CellTox dye revealed that most of the cells were permeabilized and appeared non-viable (Fig 7A). Papanicolaou staining, commonly used to detect decoy cells, followed by widefield microscopy demonstrated that detached cells had enlarged nuclei, intranuclear inclusion bodies and small and irregular cytoplasms (Figs 7B and S5), reminiscent of decoy cells type 1 [38,39]. Additionally, most decoy-like cells had intact nuclei and some cells displayed membrane ballooning (S5 Fig). Immunofluorescence staining revealed that the majority of decoy-like cells expressed agnoprotein and Vp1 (Fig 7C), and immunoblot confirmed the expression of viral proteins (Fig 7D). Next, we examined if decoy-like cells could transmit BKPyV. Cells were washed and pelleted before inoculation onto RPTECs. As a control, we included the last wash-supernatant. Immunofluorescence staining demonstrated that infection with decoy-like cells yielded strikingly more infected cells compared to the control supernatant (Fig 7E), demonstrating that the decoy-like cells harbor infectious virus.

We next investigated the detachment of BKPyV infected RPTECs by live-cell imaging in the presence of CellTox dye and the membrane stain CellMask. Due to focus-distance limitations, we used confluent RPTEC-monolayers in chamberslides. Z-stacks of RPTEC-

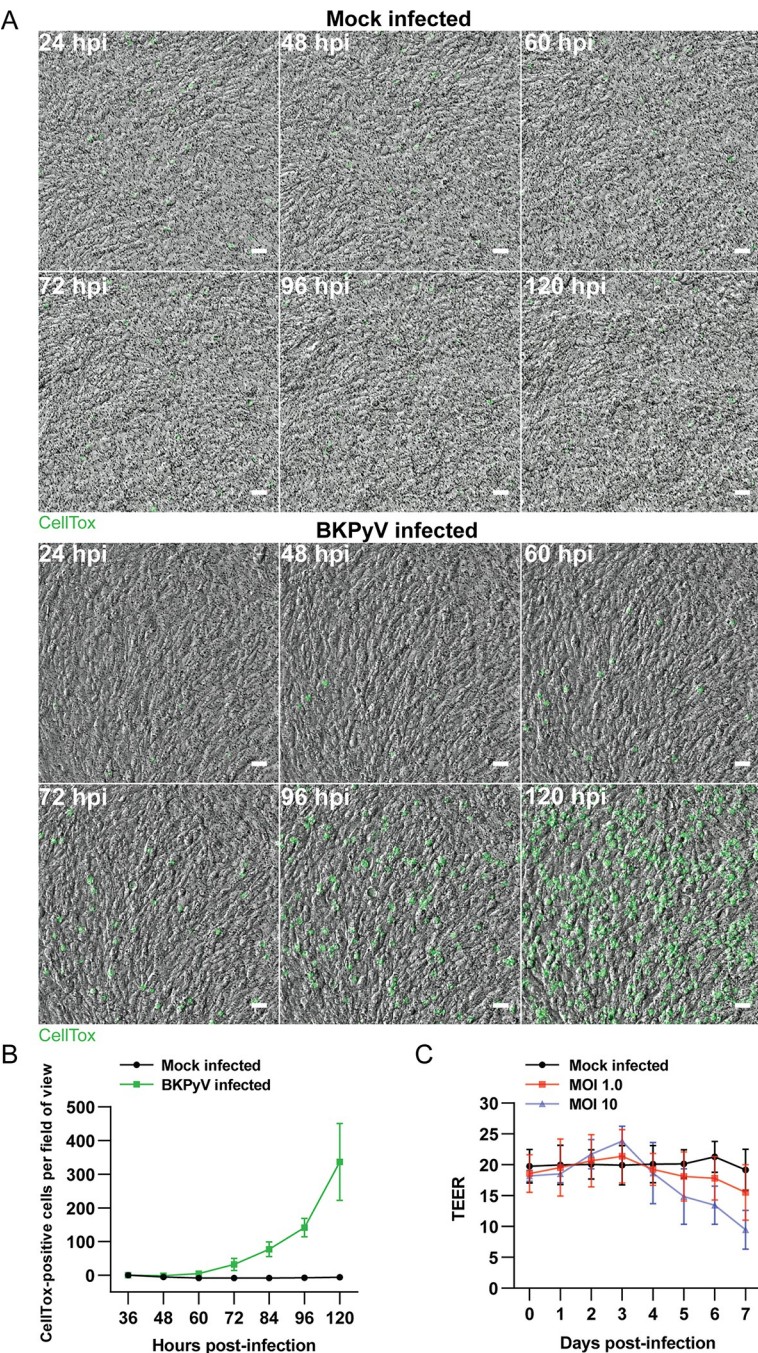

**Fig 6. Time-lapse imaging and TEER-monitoring throughout BKPyV replication.** (A) Time-lapse imaging of mock infected and BKPyV infected (MOI 1) polarized RPTECs. Nuclei of permeabilized cells are stained with CellTox dye. Representative images from two independent experiments at 24, 48, 60, 72, 96 and 120 hpi are shown. Scale bar 50 μm. (B) Quantitation of cell permeabilization in (A). Presented is the number of new CellTox-positive cells from 36 hpi. Error bars represent ± SD, n = 2. (C) TEER-values of mock infected and BKPyV infected (MOI 0.1 and 10) polarized RPTECs from 0 to 7 dpi. Data is derived from three to six biological replicates and error bars represent ± SD.

monolayers at 4 dpi revealed that CellTox-fluorescent cells were localized 10 to 20 μm above the non-permeabilized monolayer, indicating that permeabilized cells detached from the monolayer (S4 Fig).

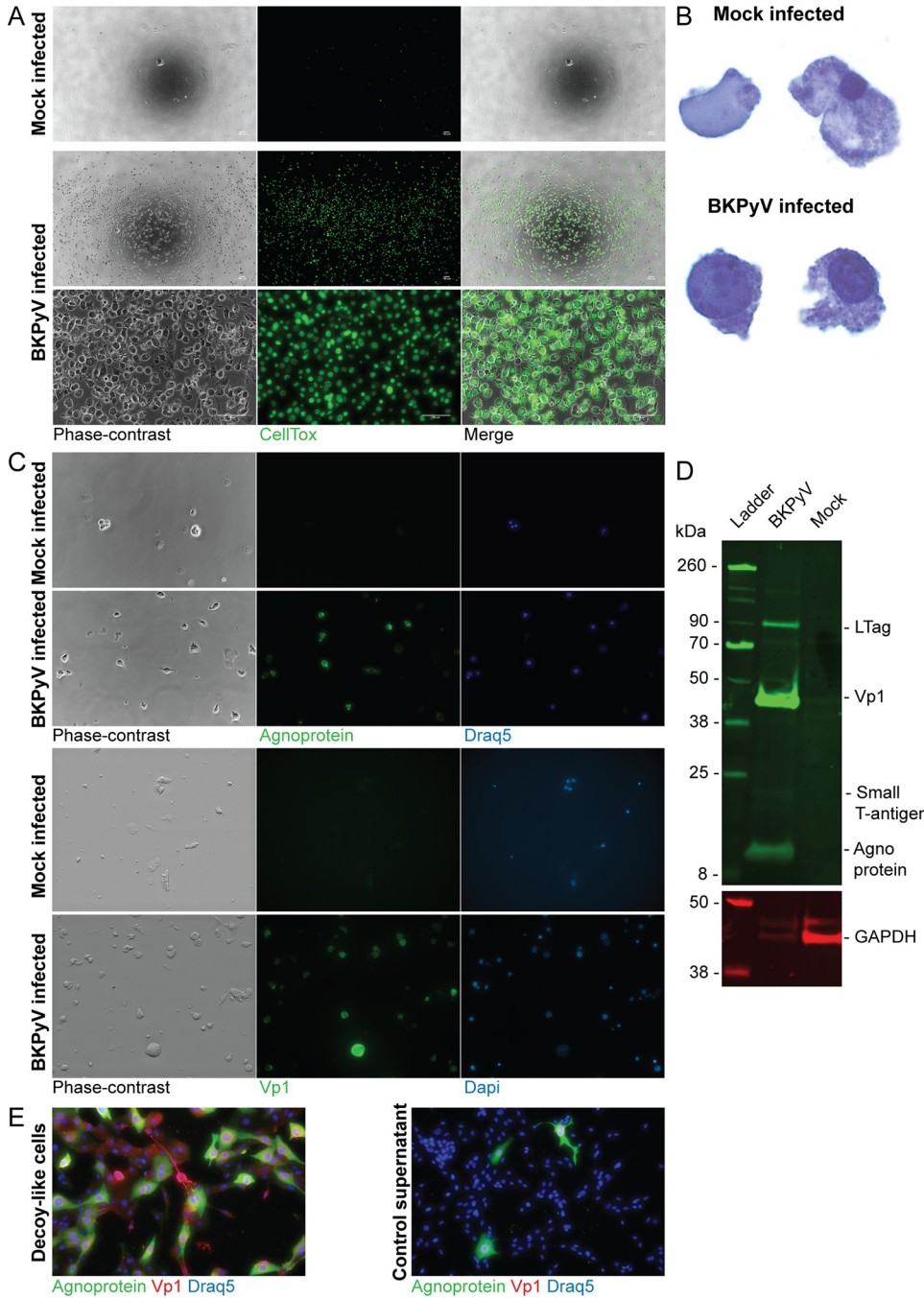

**Fig 7. BKPyV replication leads to shedding of infectious decoy cells.** Widefield microscopy of detached cells harvested from supernatants of mock infected and BKPyV infected (MOI 1) polarized RPTECs at 5 dpi. Harvested cells were imaged with phase-contrast and fluorescence microscopy. All images are representative images from two independent experiments, except (B) which is derived from one experiment. (A) Detached cells incubated with CellTox dye (green) and imaged with a combination of phase-contrast and fluorescence microscopy. Scale bar 100 µm. (B) Detached cells after Papanicolaou staining. (C) Immunofluorescence staining of detached cells from mock infected and BKPyV infected (MOI 1) polarized RPTECs using rabbit serums against Vp1 and agnoprotein. (D) Western blot of lysates of detached RPTECs harvested at 5 dpi from BKPyV infected (MOI 1) RPTECs. Mock infected RPTEC-lysate was used as the negative control. The membrane was probed with rabbit serums against N-terminal LTag, Vp1 and agnoprotein and an antibody against GAPDH. A representative blot from two experiments is shown. (E) Immunofluorescence staining of non-polarized RPTECs after infection with decoy cells harvested from BKPyV infected polarized RPTECs. A rabbit serum against agnoprotein (green) and an antibody against Vp1 (4942) (red) were used. Prior to infection, the decoy cells were washed and centrifuged five times. The supernatant from the last centrifugation was used as a control. Images are representative images from two independents experiments.

Epithelial tissues have a special mechanism, denoted extrusion, for removal of dead or unwanted cells whilst maintaining the integrity of the epithelial barrier [40]. Besides, some viruses have been reported to trigger extrusion in intestinal and airway epithelium [36,37]. To investigate if BKPyV infected RPTECs undergo extrusion, we performed time-lapse imaging with CellTox and CellMask dye [41], to visualize detachment at the single-cell level. We observed that most detaching cells were permeabilized and appeared to undergo extrusion as demonstrated by compression and subsequent upward migration of the permeabilized cell while the confluent monolayer was maintained (Fig 8A and S3 Video). However, some cells sloughed off the surface, leaving a hole in the monolayer (Fig 8B and S4 Video). Quantification of cell fate revealed that over 80% of the CellTox-fluorescent cells were detached (Fig 8C) and 70% of detached cells appeared to undergo extrusion while 30% sloughed off (Fig 8D).

We conclude that BKPyV infection induces cell death and subsequent detachment into the apical compartment. The detached cells resembled decoy cells and could transmit BKPyV. Extrusion of dead cells seems to preserve the epithelial barrier integrity.

## Neutralizing antibodies inhibit BKPyV spread in polarized cell layers

Antibodies have been shown to undergo transepithelial transport across polarized cells and inhibit viral infection [42–44]. To investigate if BKPyV-specific antibodies could undergo transepithelial transport and inhibit *de novo* BKPyV infection, we infected polarized RPTECs (MOI 0.1) and added a neutralizing BKPyV-specific Vp1-antibody in the basolateral compartment at 24 hpi. Immunofluorescence staining at 120 hpi revealed about 35% fewer infected cells compared to control inserts where a non-neutralizing antibody was added (Fig 8E).

We conclude that neutralizing antibodies traverse the tight RPTEC-layer, possibly by transcytosis, and inhibit spread of BKPyV infection.

## Discussion

Current *in vitro* studies on BKPyV replication have been performed in non-polarized cell cultures. In this study we established a polarized human renal epithelial cell model by culturing primary RPTECs on permeable inserts. Using this model, we demonstrate that BKPyV preferentially enters RPTECs via the apical membrane and that BKPyV replication results in lytic release and shedding of decoy-like cells. As cell shedding mainly occurs by extrusion, the epithelial barrier is maintained for some time, retaining viral progeny in the apical compartment. However, high-level BKPyV replication is gradually damaging the barrier, allowing virus and viral DNA to leak into the basolateral compartment. Basolateral addition of BKPyV-specific neutralizing antibodies inhibited *de novo* infections. BKPyV replication in our model closely emulates BKPyV replication in tubule epithelial cells *in vivo* and gives new insight into BKPyV reno-urinary dissemination and cytopathology.

BKPyV preferentially entered the cells via the apical membrane. Directional virus entry in epithelial cells depends on receptor distribution [18]. For instance, entry of the closely related polyomavirus SV40 and of rotavirus is suggested to be apical due to apical distribution of entry receptors [45–47]. In non-polarized cells, BKPyV has been shown to use gangliosides [27,28] and a N-linked glycoprotein containing $\alpha(2,3)$-linked sialic acid [29] as receptors. In polarized cells, gangliosides and sialic acids are known to be asymmetrically distributed to the apical membrane [47–50]. In agreement with this, WGA-staining showed more sialic acids on the apical than the basolateral membrane. Moreover, neuraminidase pre-treatment reduced BKPyV infectivity, demonstrating that BKPyV infection of polarized RPTECs rely on sialic acids on the apical membrane.

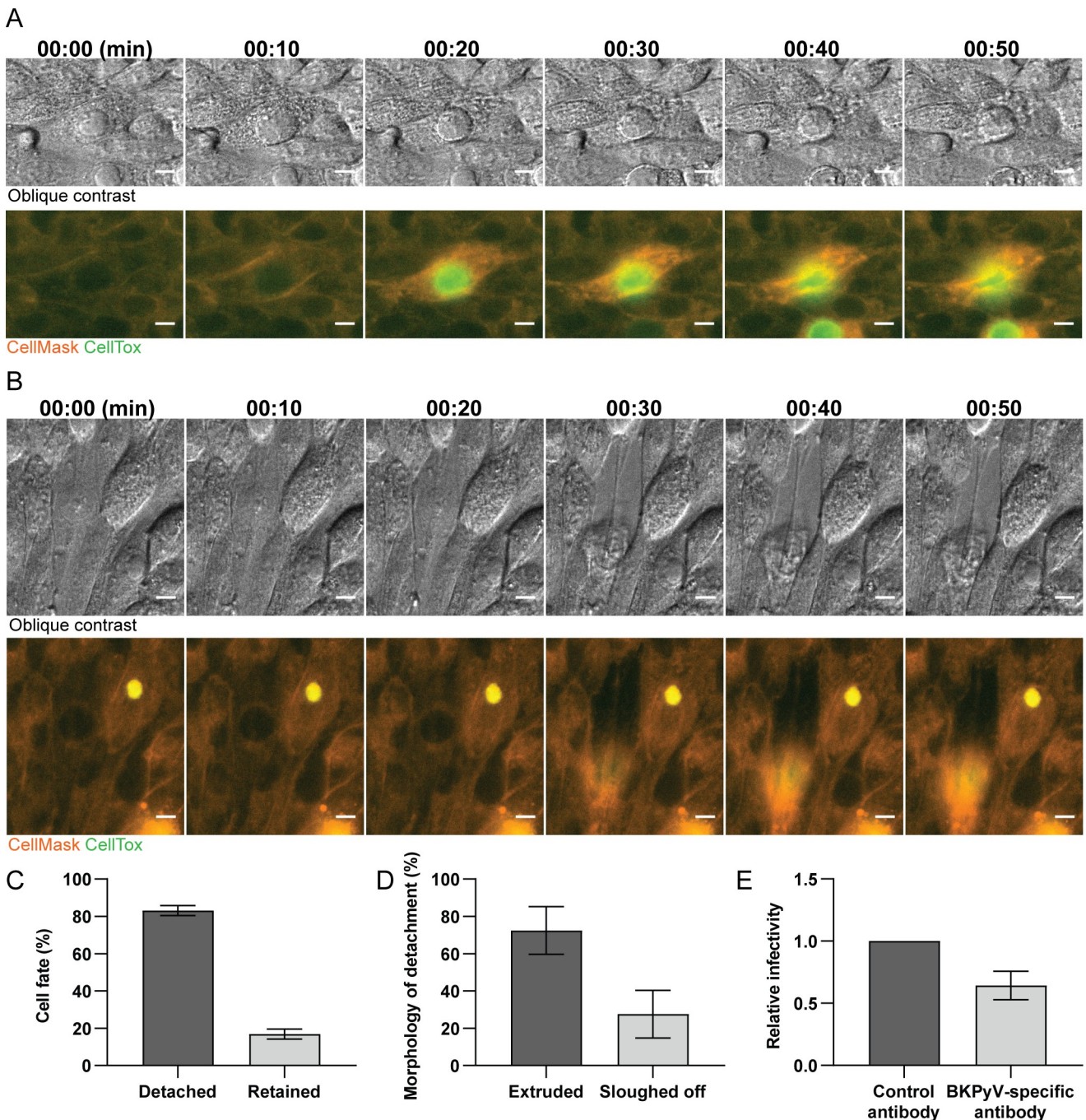

**Fig 8. Cell shedding mainly occurs via extrusion.** (A and B) Representative time-lapses with oblique contrast and fluorescence microscopy of a cell undergoing extrusion (A) or sloughing off (B). Membranes are stained with CellMask (orange) and the nuclei of permeabilized cells are stained with CellTox dye (green). Scale bar 10 μm. Images are representative time-lapses from two independent experiments. (C) Quantification of cell fate of CellTox-positive cells at 4 dpi. Data is derived from two independent experiments where 653 cells have been examined. Error bars represent ± SD. (D) Classification of the morphology of detachment for CellTox-positive cells at 4 dpi (extrusion vs. slough off). Data is derived from two independent experiments and 71 examined cells. Error bars represent ± SD. (E) Neutralizing antibodies undergo transepithelial transport and inhibit spread of BKPyV infection. Polarized RPTECs were infected with a low MOI (0.1) and at 24 hpi, a BKPyV-specific neutralizing antibody or control antibody was added to the basolateral compartment. At 5 dpi, cells were stained for LTag with a mouse antibody (Pab416) and a C-terminal LTag rabbit serum and the number of infected cells were counted. Data is derived from two independent experiments and error bars represent ± SD.

The primary mode of progeny release for non-enveloped DNA viruses is considered to be host cell lysis [51]. For BKPyV, this is supported by studies of renal allograft biopsies from PyVAN patients [10,11,33]. On the other hand, non-lytic release has been suggested for BKPyV [30,52] as well as for SV40 [53]. When we measured cell lysis by CellTox-fluorescence and LDH release, no increase in lysis was detected up to 72 hpi, a timepoint when considerable progeny release into the apical compartment had taken place. However, live-cell imaging of single cells at 72 hpi did reveal a small increase in dead cells, suggesting that progeny was indeed released by host cell lysis. Although host cell lysis seems to be the most important mechanism for BKPyV release from polarized RPTECs, we cannot exclude a parallel non-lytic release mechanism.

Even though lytic virus release normally is considered non-directional [47], several factors may explain why BKPyV was exclusively detected in the apical compartment up to 120 hpi. Firstly, since the cells were cultured in the apical compartment, lysis released BKPyV into the apical compartment. Secondly, due to shedding of infected cells by extrusion, the cell layer was intact until 5 dpi, preventing released BKPyV from leaking into the basolateral compartment. Of note, this is the first time that shedding of BKPyV infected epithelial cells by extrusion is suggested. Third, released virions presumably bound to the apical membranes, further hindering virus from traversing the insert. The two latter points are supported by the virus release assay (Figs 4A and S3A), which demonstrates that with an intact cell layer, only miniscule amounts of the inoculating virus can cross from the apical to the basolateral compartment. Lastly, apically shed cells harbored virus and contributed to the high apical BKPyV load.

How do our results fit with *in vivo* findings and increase our understanding of BKPyV dissemination and cytopathology in the reno-urinary tract? BKPyV infected polarized RPTECs showcased cytopathic changes described in renal allografts from PyVAN patients. These were enlarged nuclei with inclusions, cell rounding, cytoplasmic vacuolization, cell lysis and detachment [10,11,14,54]. Furthermore, our finding of preferential BKPyV release into the apical compartment agrees with observations in PyVAN patients, as they always have higher urinary- than plasma BKPyV load and develop viruria prior to DNAemia [12,55,56]. Apical entry also fits well with the suggested importance of ureteric reflux for multi-site spread of BKPyV in the allograft [12].

Despite BKPyV replication causing more than a 100-fold increase in viral progeny in the apical compartment and abundant cell shedding, the epithelial barrier seemed intact until 5 dpi, presumably due to extrusion of lysed BKPyV infected cells. Based on these findings, we propose that during low-level BKPyV replication in the kidneys of immunocompetent individuals or renal allografts of kidney transplant recipients, progeny virus is mainly released into the tubular lumen and disseminate intra-luminally along the tubular system. Importantly, extrusion of BKPyV infected cells keeps the epithelial lining of the tubular lumen intact. In immunocompetent individuals, we expect that infected tubule epithelial cells will eventually interact with the immune system [57] and viral replication will be inhibited.

After five days of high-level BKPyV replication in polarized RPTECs, the number of decoy-like cells increased, the epithelial barrier was disrupted and BKPyV leaked into the basolateral compartment. This suggests that during unrestricted high-level replication of donor-derived BKPyV in the kidney allograft [58], widespread cell lysis will eventually disrupt the epithelial barrier. BKPyV and intracellular BKPyV-DNA from lysed cells will leak into the interstitial fluid and blood and an increased number of decoy cell will be found in urine, together giving the diagnosis presumptive PyVAN [5]. The decoy cells can potentially act as vessels to promote further spread of BKPyV in the reno-urinary tract.

We observed that neutralizing antibodies could cross the epithelial cell layer and reduce *de novo* BKPyV infection, suggesting that treatment with intravenous BKPyV-specific

neutralizing antibodies could be beneficial as treatment of PyVAN. However, to clear BKPyV infected cells, BKPyV-specific cytotoxic T cells are needed [59]. Currently, there are several ongoing clinical trials with BKPyV-specific neutralizing antibodies and donor-derived cytotoxic T lymphocytes [60].

To establish a persistent BKPyV infection in epithelial cells of the reno-urinary tract, BKPyV must be able to evade immune sensing. BKPyV miRNA has been reported to target the stress-induced protein ULBP3 to reduce killing by natural killer cells [61]. Furthermore, Manzetti *et al* demonstrated that BKPyV evades innate immunity by disrupting the mitochondrial network and promoting mitophagy [62]. Here we for the first time demonstrate that extrusion of BKPyV infected epithelial cells leads to containment of viral progeny in the apical compartment, which is expected to delay the contact with the immune system [63]. Together, this highlights that BKPyV utilizes multiple strategies for immune evasion.

In order to fully understand the pathogenesis of BKPyV infection, including primary infection and entry, multiplication, spread within the body, the immune response and the potential kidney damage, an animal model would be very useful. Unfortunately, this is lacking. Polarized cell culture models, such as ours, is a valuable alternative tool to study BKPyV under conditions that more closely reflects the renal epithelium and tubular system than traditional cell cultures.

Taken together, we utilized a novel cell model of polarized renal tubule epithelial cells to characterize local BKPyV dissemination and cytopathological changes associated with BKPyV infection. Using this model, we establish a preferential apical entry of BKPyV, a predominant release of viral progeny into the apical compartment via host cell lysis and decoy cell shedding and finally that extrusion contain BKPyV in the apical compartment, suggesting that BKPyV *in vivo* spreads intra-luminally along the nephron to the pelvis and bladder and thereby delay immune detection.

## Materials and methods

### Cells and virus

Primary human RPTECs (Lonza) were cultured in renal epithelial growth medium (REGM; Lonza) containing 0.5% fetal bovine serum in a humidified 5% $CO_2$ incubator at 37˚C. Cesium-chloride gradient purified BKPyV Dunlop was used for all infections.

### Culture and infection of polarized RPTECs

For polarization, RPTECs were seeded on collagen-coated (recombinant human collagen type I; Sigma-Aldrich) permeable polyester-membrane cell culture inserts and cultured for 8 to 15 days. The following inserts were used: Falcon inserts with pore size 1.0 μm or 3.0 μm and Corning Transwell-inserts with pore size 0.4 μm, all with 0.3 $cm^2$ growth area.

Polarized RPTECs were infected with an inoculum of 100 μl. Apical infection was done by adding inoculum inside the insert. Basolateral infection was performed by temporarily inverting inserts and adding inoculum on top. Infections were performed for 2 hours at 37˚C.

For neuraminidase-experiments, cells were pretreated with 100 mU/ml neuraminidase type V from clostridium perfringens (Merck) for 1 hour prior to apical infection.

For neutralization experiments, polarized RPTECs were apically infected (MOI 0.1). At 24 hpi, a mouse monoclonal BKPyV-specific antibody (Virostat 4942) or control antibody (Virostat 4944) was added to the basolateral compartment at a concentration of 7 μg/ml. At 5 dpi, the number of infected cells was determined with immunofluorescence staining.

## Immunofluorescence staining, microscopy and image analysis

Cells were fixed in ice-cold methanol or 4% paraformaldehyde for 10–15 minutes. PFA-fixed cells were permeabilized with permeabilization buffer (DPBS with 1% BSA and 0.1% Triton-X100) for 10 minutes. Immunofluorescence staining was performed as previously described [64] except that primary and secondary antibody staining was performed for 1 hour at room temperature. The following antibodies and serums were used to stain for viral proteins:, mouse anti-BKPyV Vp1 (4942; 2.8 ug/ml; Virostat), mouse anti-LTag (Pab416; 1:100; Merck Millipore), rabbit serums against BKPyV agnoprotein (1:1000) [65], BKPyV N-terminal LTag (1:1000) [66,67], BKPyV C-terminal LTag (1:1000) [67] and BKPyV Vp1 (1:1000) [68]. The following antibodies were used to stain for cellular proteins: rabbit monoclonal anti-Na/K-ATPase (ab76020; 1:500; Abcam), rabbit polyclonal anti-ZO-1 (61–7300; 1:100; Invitrogen) and mouse monoclonal anti-acetylated α-tubulin (sc-23950; 1:100; SCBT). Secondary antibodies used were goat anti-rabbit Alexa Fluor 488 and anti-mouse Alexa Fluor 568 (Invitrogen). Staining with Texas Red-conjugated WGA (10 µg/ml; Thermo Fisher) was performed for 30 minutes on living cells. Insert-membranes was cut out and mounted on a cover glass with mowiol or ProLong Diamond mounting medium.

Widefield microscopy was performed using a Nikon TE2000-microscope with a 4x (NA 0.13) and 20x objective (NA 0.45) and NIS Elements Basic Research software. Confocal microscopy was performed using a Zeiss LSM800 confocal microscope with a 40x water-objective (NA 1.2) and Zeiss ZEN blue software. All images were processed with FIJI/ImageJ.

To measure Vp1-staining, z-stacks of equal size were acquired. Identical acquisition settings were used for each replicate. Using ImageJ, sum z-projections were generated and mean fluorescence was measured. The mean fluorescence in z-projections from mock infected inserts were subtracted as background.

## Transepithelial resistance

TEER was measured using a Millicell ERS-2 voltohmmeter with an adjustable electrode (STX02) in a 12-well plate. TEER (Ohm $*$ cm$^2$) was calculated by subtracting the background of an empty insert from the average TEER and multiplying the value by the surface area (0.3 cm$^2$). All measurements were done in duplicate or triplicate.

## FITC-Dextran diffusion assay

The assay was adapted from two publications [69,70]. Polarized RPTECs were incubated with 100 µl of 0.1 mg/ml FITC-Dextran MW 20k (Merck) in the apical compartment for 1 hour at 37˚C. Supernatant was harvested from the basolateral compartment and fluorescence was measured using a Tecan Infinite 200 Pro plate reader with an excitation wavelength of 488 nm and emission wavelength of 518 nm. All measurements were performed in duplicate or triplicate.

## P-glycoprotein assay

The assay was adapted from the following studies [26,69,71]. Polarized RPTECs were pre-treated with 5 µM of the P-gp inhibitor PSC-833 (Merck) or solvent (0.125% DMSO), followed by addition of REGM with 1 µM Calcein-AM (Invitrogen) and 5 µM PSC-833 or solvent. After incubation at 37˚C for 1 hour, cells were washed twice with DPBS and lysed with 1% Triton X-100. Fluorescence in each sample was measured using a plate reader as described for the FITC-dextran diffusion assay.

## Transmission electron microscopy

The protocol was adapted from two reports [72,73]. Samples were fixed in 0.5% glutaraldehyde and 4% formaldehyde in PHEM-buffer (60 mM PIPES, 25 mM HEPES, 10 mM EGTA, 4 mM MgSO$_4$·7H$_2$O) for 30 minutes before fixing again with 4% formaldehyde, 0.5% glutaraldehyde, and 0.05% malachite green in PHEM-buffer (2 min vacuum on-off-on-off-on-off-on, 100 W) using a Ted Pella microwave processor. Samples were post-fixed with 1% osmium tetroxide, 1% K$_3$Fe(CN)$_6$ in 0.1 M cacodylic acid buffer, post-stained with 1% tannic acid and 1% uranyl acetate and dehydrated in increasing ethanol series before embedding in an Epon-equivalent. 70 nm sections were imaged using a Hitachi HT7800 transmission electron microscope with a Xarosa-camera.

## Immunoblots

Immunoblot was performed as previously described [74] except Halt protease- and phosphatase-inhibitor was used in the lysis buffer, lysates were pretreated with Pierce nuclease and membranes were blocked with LI-COR Intercept TBS blocking buffer. The following primary antibody and serums were used: rabbit serums against BKPyV Vp1 (1:10 000) [68], BKPyV N-terminal LTag (1:2000) [67], BKPyV agnoprotein (1:10 000) [65] and mouse anti-GAPDH (ab8245; 1:2000; Abcam). The secondary antibodies used were goat anti-rabbit 800CW and goat anti-mouse 680RD from LI-COR Biosciences. Detection was done with the Odyssey CLx imaging system and Image studio.

## Virus diffusion assay

Cell-free collagen-coated Transwell-inserts (pore size 0.4 μm) and Falcon-inserts (pore size 1.0 μm) were incubated with 100 μl of CsCl-purified BKPyV in REGM (250 000–500 000 fluorescent focus units (FFU)) inside the insert for two hours at 37˚C. The basal medium was then harvested and inoculated on non-polarized RPTECs. Infectivity was determined by immunofluorescent staining for agnoprotein and Vp1 (4942) at 3 dpi. To calculate the percentage of virus that diffused across the insert, the infectivity of the basal medium and the initial virus suspension was compared.

## Virus release assay

Polarized RPTECs on Falcon-inserts were apically infected before supernatants were harvested at the indicated timepoints and analyzed for BKPyV-DNA or infectious BKPyV. BKPyV-DNA was quantitated by a BKPyV-specific qPCR targeting the BKPyV LTag gene [75]. Infectivity was measured by inoculating diluted supernatants onto non-polarized RPTECs. At 3 dpi, immunofluorescence staining against agnoprotein and Vp1 (4942) was performed and infected cells were counted using the object count feature of the NIS Elements basic research software. All replicates were performed in duplicate.

## Cell viability

Cell viability and CPE were examined by phase-contrast microscopy, by LDH release (Promega LDH-Glo Cytotoxicity assay according to the manufacturer's instructions) and by the use of the plasma membrane impermeable CellTox dye (x1) from the Promega CellTox Green cytotoxicity assay. For the CellTox-experiments, CellTox dye was added to the medium before images were acquired. ImageJ was used to measure the total fluorescence per image. Each measurement was performed in duplicate.

## Live-cell imaging

Live-cell imaging of inserts were done with Falcon-inserts in a 24-well plate with confocal glass bottom (CellVis P24-1.5H-N) in REGM with CellTox dye (1x) and Hoechst (0,1 μg/ml). Images were acquired using an automated Zeiss CellDiscoverer 7 microscope with a 5x objective (NA 0.35) and 2x tube lens at 37˚C with humidity and 5% $CO_2$.

To visualize detachment, RPTECs were cultured in fibronectin-coated Lab-Tek 8-well chamberslides (#1 coverglass). RPTECs were infected (MOI 1) when fully confluent monolayers had formed. At 4 dpi, CellMask (0.5x) and CellTox dye (1x) were added, and live-cell imaging was performed as above, but with a 20x objective (NA 0.95) and 2x tube lens. Images were acquired every 10 minutes.

CellTox-fluorescent cells were classified as detached if the nucleus was out of focus or the cell was floating on top of the monolayer, while cells that were adherent and had the nucleus in the same focus plane as the monolayer were classified as retained. Cells that detached without leaving a gap in the monolayer were classified as extruded while cells that left a gap were classified as sloughed off.

## Harvest and examination of detached cells

Detached cells were harvested by gently aspirating the apical supernatant, washing once and then pooling supernatants and wash-medium. For immunofluorescence or standard Papanicolaou staining, cells were fixed with ThinPrep PreservCyt-solution and processed with the ThinPrep 5000 Processor before staining. For immunoblot, cells were processed identically as RPTEC lysates. For widefield microscopy, supernatants were transferred to wells to sediment followed by imaging with a Nikon TE2000-microscope (20x objective) or a Zeiss CellDiscoverer 7 microscope (20x objective, 2x tubelens). To examine the infectivity of detached cells, detached cells were washed and centrifuged five times to remove extracellular virus before the cells were inoculated onto non-polarized RPTECs. Wash supernatant was used as a control.

## Supporting information

**S1 Data. Excel spreadsheet containing the numerical values used for graphs and statistical analysis for figure panels 1D, 1E, 1F, 3A, 3B, 3D, 3E, 4A, 4B, 5B, 5C, 6B, 6C, 8C, 8D, 8E, S1E, S3A, S3B, S3C, S3D and S3E.**
(XLSX)

**S1 Fig. RPTECs grown on cell culture inserts with 0.4 μm and 3.0 μm pore size.** Immunofluorescence staining of RPTECs at 10 dps on Transwell-inserts, pore size 0.4 μm, against markers of apico-basal polarity: (A) Na/K-ATPase (green), (B) acetylated α-tubulin (red) and (C) ZO-1 (green). Nuclei were stained with Draq5 (blue). Images are representative images from at least three independent experiments. (D) Diffusion of FITC-dextran across polarized and non-polarized RPTECs. Data is normalized to the non-polarized control, n = 7 and error bars represent ± SD. *** = P < 0.001, one sample $t$ test. (E) Accumulation of intracellular calcein AM with or without Psc-833, quantified by measuring intracellular fluorescence with a plate reader. Data is normalized to the untreated control. Error bars represent ± SD and n = 3. * = P < 0.05, one sample $t$ test. (F) Confocal microscopy of RPTECs grown on Falcon-inserts with 3.0 μm pore size at 9 dps. Cell membranes were stained with CellMask (orange) while nuclei were stained with Draq5 (blue). Representative images from a z-stack from two independent experiments.
(TIF)

**S2 Fig. Supplemental images to Fig 3.** Immunofluorescence staining against agnoprotein (green) and Vp1 (4942) (red) in polarized RPTECs (A) or non-polarized RPTECs (B), infected via the apical or basolateral compartment. (C) Immunofluorescence staining for Vp1 (4942) (red) in polarized RPTECs treated with or without neuraminidase prior to infection. Nuclei were stained with Draq5 in all images. All images are representative images from at least three independent experiments. (D) Confocal microscopy of non-polarized RPTECs stained with Texas Red conjugated wheat germ agglutinin (red) via the apical or basolateral membrane. Scale bar 10 μm.
(TIF)

**S3 Fig. Virus release and cell viability with increased MOI.** (A) Purified BKPyV was added to the apical compartment. After 2 hours, apical and basolateral supernatants were collected, BKPyV-DNA load (log10 copies/ml) was determined by qPCR and apical and basolateral distribution was calculated. Data was generated from three independent experiments and error bars represent ± SD. (B and C) BKPyV-DNA load (log10 copies/ml) in apical and basolateral supernatants collected from infected polarized RPTECs at indicated timepoints. (B) MOI 3 and (C) MOI 30. Data is generated from three independent experiments and error bars represent ± SD. (D) Widefield microscopy of uninfected and infected (MOI 3 and 30) RPTECs incubated with CellTox dye. Data is presented as CellTox-ratio (mean total fluorescence from infected inserts/mean total fluorescence from mock infected inserts). Error bars represent ± SD and n = 2. (E) Release of LDH into apical supernatants as measured by Promega LDH-Glo Cytotoxicity assay. Data is presented as LDH-ratio (infected-RLU/mock-RLU). Error bars represent ± SD and n = 2.
(TIF)

**S4 Fig. Permeabilized cells lie in a different z-plane than the viable monolayer.** Fully confluent RPTECs were BKPyV infected (MOI 1) and at 4 dpi cells were stained with CellTox dye (green) and CellMask (orange) followed by live-cell imaging and acquisition of z-stacks. Shown is oblique contrast (top row), CellTox (green, second row), CellMask (orange, third row), merge of oblique contrast and CellTox (fourth row) and merge of CellMask and CellTox (bottom row). Each column displays the same z-slice. Images are from a representative z-stack derived from two independent experiments. Scale bar 20 μm.
(TIF)

**S5 Fig. Widefield microscopy of detached cells.** Harvested cells incubated with CellTox dye (green) and Hoechst (blue) and imaged with a combination of oblique contrast and fluorescence microscopy using a Zeiss Celldiscoverer 7 with a 20x objective and 2x tube lens. Images are derived from two independent experiments. Scale bar 10 μm.
(TIF)

**S1 Video. Time-lapse microscopy of polarized RPTECs infected with BKPyV (MOI 1) from 5 to 120 hpi with image acquisition every 30 minutes.** The nuclei of permeabilized cells are stained with CellTox dye (green). Scale bar 50 μm.
(AVI)

**S2 Video. Time-lapse microscopy of mock infected polarized RPTECs from 5 to 120 hpi with image acquisition every 30 minutes.** The nuclei of permeabilized cells are stained with CellTox dye (green). Scale bar 50 μm.
(AVI)

**S3 Video. Time-lapse microscopy of cell undergoing extrusion as showcased by compression of the permeabilized cell and subsequent upwards migration of the cell.** Membranes

are stained with CellMask (orange) and the nuclei of permeabilized cells are stained with Cell-Tox dye (green). Left panel shows oblique contrast, right panel shows CellMask and CellTox. Images were acquired every 10 minutes. Scale bar 10 μm.
(AVI)

**S4 Video. Time-lapse microscopy of cell sloughing off from surface.** Membranes are stained with CellMask (orange) and the nuclei of permeabilized cells are stained with CellTox dye (green). Left panel shows oblique contrast, right panel shows CellMask and CellTox. Images were acquired every 10 minutes. Scale bar 10 μm.
(AVI)

## Acknowledgments

We thank Garth D. Tylden (University Hospital of North Norway, UNN) for helpful discussions and critical reading of the manuscript, Kristian Prydz (University of Oslo) for helpful discussions, the Advanced Microscopy Core Facility at UiT–The Arctic University of Norway for the use of instruments, Randi Olsen (UiT), Kenneth Bowitz Larsen (UiT) and Mona Antonsen (UNN) for technical assistance and reagents.

## Author Contributions

**Conceptualization:** Elias Myrvoll Lorentzen, Christine Hanssen Rinaldo.

**Data curation:** Elias Myrvoll Lorentzen.

**Formal analysis:** Elias Myrvoll Lorentzen, Stian Henriksen.

**Funding acquisition:** Elias Myrvoll Lorentzen, Christine Hanssen Rinaldo.

**Investigation:** Elias Myrvoll Lorentzen, Stian Henriksen.

**Methodology:** Elias Myrvoll Lorentzen, Stian Henriksen.

**Project administration:** Christine Hanssen Rinaldo.

**Supervision:** Christine Hanssen Rinaldo.

**Visualization:** Elias Myrvoll Lorentzen.

**Writing – original draft:** Elias Myrvoll Lorentzen.

**Writing – review & editing:** Elias Myrvoll Lorentzen, Stian Henriksen, Christine Hanssen Rinaldo.

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
