## [Decision Letter · Decision Letter 0]

8 Aug 2023

Dear Professor Rinaldo,

Thank you very much for submitting your manuscript "Modelling BK Polyomavirus dissemination and cytopathology using polarized human renal tubule epithelial cells" for consideration at PLOS Pathogens. As with all papers reviewed by the journal, your manuscript was reviewed by members of the editorial board and by several independent reviewers. The reviewers appreciated the attention to an important topic. Based on the reviews, we are likely to accept this manuscript for publication, providing that you modify the manuscript according to the review recommendations.

Sincerely,

Walter J. Atwood

Academic Editor

PLOS Pathogens

Patrick Hearing

Section Editor

PLOS Pathogens

Kasturi Haldar

Editor-in-Chief

PLOS Pathogens

orcid.org/0000-0001-5065-158X

Michael Malim

Editor-in-Chief

PLOS Pathogens

orcid.org/0000-0002-7699-2064

Reviewer Comments (if any, and for reference):

Reviewer's Responses to Questions

**Part I - Summary**

Reviewer #1: BKPyV primaryly infects renal tubular cells and has the potential to destroy the kidney in immunosuppressed individuals; this infection is particularly detrimental in renal transplant recipients. It is currently assumed that intrarenal dissemination of the virus occurs by continuity in renal tubules, however, a detail understanding of this process is not known.

In this study the BKPyV infection is evaluated in a polarized renal proximal tubule epithelial cell line that has the potential to most closely resemble the infectious process in renal tubules.

The study is novel and has experimental/methodological value. Furthermore, conclusions from this study help better understand the lag between the earlier stages of infection and the onset of clinical manifestations,

Reviewer #2: In this manuscript (PPATHOGENS-D-23-01211), Lorentzen et al describe a new cultured cell model system to study infection by the pathogenic human polyomavirus, BKPyV. BKV infection is widespread in the population and causes severe disease in immunosuppressed individuals, such as renal transplant recipients and patients undergoing bone marrow or stem cell transplants. BKV normally replicates in polarized renal tubule epithelial cells, but in vitro studies are performed in monolayer cell cultures. A few years ago, the Imperiale lab described BKV replication in cultured non-polarized renal proximal tubule epithelial cells (RPTECs). Here, the authors establish polarized cultures of RPTECs on permeable inserts, show BKV replication and cytopathic effects in infected cells, and that the virus preferentially infects and is released from the apical surface (as is the case for the closely related primate polyomavirus, SV40, which also replicates preferentially in kidney), until very late in infection. Virus is also contained in extruded cells with maintenance of epithelial integrity. The work appears competently done and this is likely to be a valuable model system to study this virus. However, the current work is largely descriptive, and they have not yet used the system to provide new mechanistic insights into BKV replication or pathogenesis.

**Part II – Major Issues: Key Experiments Required for Acceptance**

Reviewer #1: None.

Reviewer #2: 1. This manuscript would be greatly strengthened if they use their system to address unanswered questions regarding BKV replication or pathogenesis.

2. Their model for how BKV spreads in vivo is intriguing and consistent with their in vitro studies, but not directly addressed in the experiments.

**Part III – Minor Issues: Editorial and Data Presentation Modifications**

Reviewer #1: - In the materials and methods, immunofluorescence staining and microscopy section, a list of antibodies is presented without further clarification. Some of these antibodies are never mentioned in the text and are only mentioned in the figure legends. Since evaluation of the polarized renal tubular cell line is relatively novel, it would be useful if the authors could briefly explain the purpose of the stains. For example, separate stains for characterization of the viral infection from the stains done to characterize the tubular cell itself (ZO-1, tubulin, etc).

- There is a typo in the last line of page 16 (pylon should be pelvis)

Reviewer #2: They should define “decoy cells”.

The use of the word “sank” on line 210 is unclear.

PLOS authors have the option to publish the peer review history of their article (what does this mean?). If published, this will include your full peer review and any attached files.

Reviewer #1: No

Reviewer #2: No

Figure Files:

Data Requirements:

Reproducibility:

References:

---

## [Editor Report · Decision Letter 1]

17 Aug 2023

Dear Professor Rinaldo,

We are pleased to inform you that your manuscript 'Modelling BK Polyomavirus dissemination and cytopathology using polarized human renal tubule epithelial cells' has been provisionally accepted for publication in PLOS Pathogens.

Best regards,

Walter J. Atwood

Academic Editor

PLOS Pathogens

Patrick Hearing

Section Editor

PLOS Pathogens

Kasturi Haldar

Editor-in-Chief

PLOS Pathogens

orcid.org/0000-0001-5065-158X

Michael Malim

Editor-in-Chief

PLOS Pathogens

orcid.org/0000-0002-7699-2064
---

## [Editor Report · Acceptance letter]

24 Aug 2023

Dear Professor Rinaldo,

We are delighted to inform you that your manuscript, " Modelling BK Polyomavirus dissemination and cytopathology using polarized human renal tubule epithelial cells ," has been formally accepted for publication in PLOS Pathogens.

Best regards,

Kasturi Haldar

Editor-in-Chief

PLOS Pathogens

orcid.org/0000-0001-5065-158X

Michael Malim

Editor-in-Chief

PLOS Pathogens

orcid.org/0000-0002-7699-2064